# Evaluation of Nutrient Consumption for the Prevention of Chronic Diseases in Health Promotion Services: A Controlled and Randomized Community Trial to Promote Fruits and Vegetables

**DOI:** 10.3390/ijerph20136267

**Published:** 2023-06-30

**Authors:** Suellen Fabiane Campos, Mariana Souza Lopes, Luana Caroline dos Santos, Patrícia Pinheiro de Freitas, Aline Cristine Souza Lopes

**Affiliations:** 1Department of Nutrition, Nursing School, Federal University of Minas Gerais, Belo Horizonte 30130-100, Brazil; susufabi@gmail.com (S.F.C.); luanacstos@gmail.com (L.C.d.S.); patpfreitas@gmail.com (P.P.d.F.); 2Department of Nutrition, Health Science Center, Federal University of Paraíba, João Pessoa 58051-900, Brazil; marianalopes.ufpb@gmail.com

**Keywords:** primary health care, health services, nutritional intervention, nutrients, health promotion

## Abstract

To evaluate the effectiveness of a collective intervention to encourage the consumption of fruits and vegetables on the nutrients intake for the prevention of chronic non-communicable diseases (NCDs), a randomized controlled community trial was conducted with a representative sample from the Health Academy Program. While the individuals in the control group (CG) participated in regular physical exercise, those in the intervention group (IG) also participated in a collective intervention. After 12 months, IG and CG showed a reduction in energy, omega 3 and sodium intake and an increase in the consumption of carbohydrates, fiber, vitamins, and minerals. Individuals from the CG showed an increase in phosphorus consumption and, in the IG, a reduction in the consumption of total, saturated, and polyunsaturated fats as well as an increase in the consumption of monounsaturated fats was seen. In both groups, there was an increase in the prevalence of adequate nutrients. Participation in the nutritional intervention was associated with lower consumption of energy and protein in the diet. The results pointed to the importance of individuals’ participation in the program, which, associated with nutritional intervention, promoted an improvement in the nutrient profile of the diet and the prevention and control of NCDs.

## 1. Introduction

Chronic non-communicable diseases (NCDs) are an important public health problem, accounting for more than 70% of all deaths worldwide [1]. A healthy diet has the potential to protect health and reduce the risk of NCDs. In this sense, it contributes to achieving a balance in energy intake and controlling the consumption of total fat, free sugars, and sodium by including unprocessed foods, such as fruits and vegetables (FV), and minimally processed foods, such as leguminous, oilseeds, and whole grain foods [2]. Corroborating these recommendations, a systematic review of the literature identified that the low energy content of the diet and the abundance of nutrients, such as vitamins and fibers from FV, contribute to the prevention of obesity and cardiovascular diseases [3].

In view of this, different nations, including Brazil, have sought to encourage their populations to adopt adequate and healthy diets, in addition to other healthy lifestyle factors [2,4,5,6]. In Brazil, the Food Guide for the Brazilian Population stands out in this regard when it recommends a golden rule: “Prefer unprocessed, minimally processed foods and home prepared food to ultra-processed foods” [4]. This guideline aims to encourage a diet capable of offering the nutrients needed to maintain health, focusing on food, ways of eating, and the cultural and social dimensions of eating practices [4]. In this sense, the Food Guide has been instrumental in guiding the planning of actions to promote adequate and healthy food within the health system, including health services, such as the Health Academy Program (Programa Academia da Saúde, in Portuguese—PAS). 

The PAS is a Brazilian primary health care (PHC) service which aims to contribute to the promotion and production of health care and healthy lifestyles for the population [7]. It includes in its scope the regular practice of physical activity and food and nutrition actions, among other activities [8], in order to also contribute to the prevention and control of NCDs. Studies carried out with PAS users showed that different nutritional interventions, using the Transtheoretical Model, were able to promote a reduction in the consumption of energy and foods rich in fat [9], as well as increased consumption of FV among those with lower intake of these foods [10]. 

Despite the positive results presented by the nutritional interventions developed within the scope of the PAS, it is not known whether actions to encourage the consumption of FV can also bring positive changes in the nutrient profile of the diet, especially for those nutrients aimed at the prevention and control of NCDs [5]. A systematic review of the literature suggested that nutritional interventions developed in PHC may have effects on the eating behavior of adults, although more studies are needed to assess the consistency and effectiveness of these actions in the health outcomes associated with NCDs [11]. 

Therefore, the objective of this study was to evaluate the effectiveness of a collective intervention to encourage the consumption of FV on the nutrients intake, especially the ones focused on the prevention and control of NCDs, in those who attend the Brazilian PHC health promotion service.

## 2. Methods

### 2.1. Type and Place of Study

This is a randomized controlled community trial, carried out between 2013 and 2015, with the objective of evaluating the impact of an intervention to encourage the consumption of FV in PAS users in the city of Belo Horizonte, Minas Gerais, Brazil [12]. This municipality, divided into nine administrative regions, has an estimated population of 2,521,564 inhabitants, and is the sixth largest city in the country [13]. 

Access to the PAS in Belo Horizonte occurs by spontaneous demand or by referral from a health professional [14]. The program provided free exercise sessions of physical exercises, guided by physical education professionals, three times a week, lasting 60 min, in addition to health education actions, most of which related to adequate and healthy nutrition [14,15]. Currently, there are 78 PAS units [15] distributed in nine health districts and located primarily in areas of high and very high vulnerability, according to the Health Vulnerability Index (Índice de Vulnerabilidade à Saúde, in Portuguese—IVS). This index includes socioeconomic and environmental variables to assess the degree of vulnerability to health in different areas of the municipality, classifying them as low, medium, high, or very high risk [16]. 

### 2.2. Study Sample

The selection of PAS units participating in the community trial was based on simple cluster sampling, stratified by the nine administrative regions of the municipality. At the time of the sampling process (2012), there were 50 PAS units in operation, 42 of which were eligible because they operate in the morning and are located in areas with medium and high/very high IVS; they have the predominant characteristics of the health service in the municipality, and have not participated in a nutritional intervention study in the last two years [12]. 

The eligibility criteria for the PAS units were morning operation, having medium, high or very high IVS, being in operation in the years of the sample period, and not having been the subject of research on food and nutrition in the 2 years before this research was carried out [12]. Eighteen units were randomly selected via simple random sampling, two per region: one allocated as an intervention group (IG) and another as a control group (CG). For that, the units were separated by region and, later, numbered for the draw. When similarity in IVS classification was not observed between the units sampled in the region, a new draw was conducted to identify a replacement. This sample was representative of the PAS of the municipality with medium and high/very high IVS, with 95% confidence, and an error rate of 1.4% [12]. 

In the sampled PAS units, all individuals aged 20 years or older, healthy or with a chronic disease, such as hypertension, diabetes, among others, who were regular participants in the activities offered by the health service (present according to the attendance list for the month prior to data collection) and who agreed to participate in the study, were eligible for the study. The exclusion criteria were pregnant women and people with some cognitive impairment who could not provide adequate answers the questionnaire. Of the 3763 eligible individuals, 112 (3%) did not meet the inclusion criteria and 237 (6.3%) refused to participate in the research; the final sample consisted of 3414 individuals, 1428 (43.4%) of whom were participants in the IG and 1931 (56.6%) in the CG (Figure 1).

### 2.3. Data Collection

Data collection was carried out face-to-face by students from the –dietitian course and health professionals, previously trained and accompanied by a general field supervisor and the main researcher. This training was carried out every six months and a field manual was created to support data collection and standardize the collection of variables [12].

The baseline was performed from February 2013 to June 2014; the nutritional intervention, lasting 7 months, was conducted between August 2013 and December 2014; and the reassessment was carried out 12 months after the baseline (March 2014 to March 2015).

At baseline, sociodemographic data (sex, age, marital status, per capita family income, and years of education), health data (self-perception of health, practice of physical activity, reported morbidity: arterial hypertension and diabetes mellitus), food consumption (24 h food recall—R24h), time of participation in the PAS (number of months that the individual had attended the health service by consulting the health service spreadsheet), and nutritional status (body mass index—BMI) were collected. 

For the calculation of the BMI [(BMI = weight(kg)/height(m)^2^], the weight was obtained via a single measurement on a Marte^®^ brand digital scale, model PP 180, with a capacity of 180 kg and precision of 100 g. Height was also verified via a single measurement using a portable stadiometer, brand Alturexata^®^, with a capacity of 220 cm and accuracy of 0.5 cm. Individuals with a BMI ≥ 30 kg/m^2^ were classified as having obesity [17].

Food consumption was investigated by the average of two R24h applied on different non-consecutive days, or by just one R24h in cases where the second day was not answered [12]. At baseline, 6% (*n* = 206) responded to only one R24h and at reassessment, only 3% (*n* = 68) responded. To make it easier for individuals to report and minimize portion size estimation errors, a home measurement kit was used, containing household items commonly used by the population [12,18,19].

The homemade measures of the food and reported recipes were transformed into grams or milliliters using tables and manuals to assess food consumption [20,21,22], and food labels and measurements performed by the research team (weighing and standardization) [23,24]. Then, the R24h data were tabulated in a specific program, used in a national food survey, by a trained and supervised team [23,24]. To record foods consumed in the program’s database, the content of added sugar in beverages was considered, if the individual reported adding only sugar (10% of the volume) or adding sugar and artificial sweeteners (5% of the volume) [22,25]. The addition of salt and oil was also considered in the reported home prepared food. 

For the transformation of food into nutrients, the food database and the respective food amounts were associated with the nutritional composition table prepared by the Family Budget Survey [21], with the addition of foods entered by the research group.

### 2.4. Intervention to Encourage the Consumption of Fruits and Vegetables

The individuals in the CG participated in the routine activities of the health service, which consisted of regular guided practice of physical exercises, three times a week, lasting one hour, in addition to specific health education actions carried out by PHC professionals, except for those related to FV consumption. IG participants additionally participated in a collective intervention to encourage the consumption of FV that lasted seven months. The intervention was planned and developed by a multidisciplinary team (nutritionists, psychologist, and educator) with experience in health education and food and nutrition education, and applied by nutritionists and psychologists, supported by nutrition students, under the supervision of the main researcher [26]. 

Nutritional intervention was based on the Transtheoretical Model and on the dialogic and problematizing pedagogy proposed by Paulo Freire [26]. The Transtheoretical Model has four pillars: stages of change, processes of change, self-efficacy, and decision balance [27]. There are five stages of change that assess the readiness for change: pre-contemplation, contemplation, preparation or decision, action, and maintenance. Individuals in pre-contemplation and contemplation are not considered ready to make changes in the foreseeable future. On the other hand, those in the preparation, action, and maintenance stages are, respectively, ready for changes in the next 30 days, ready for immediate changes, or have already implemented changes for more than six months.

Change processes, on the other hand, establish the understanding of how change occurs at different stages, and it can be cognitive, more focused on the initial stages of change; or behavioral, better serving the final stages of change [27,28,29,30]. Self-efficacy, in turn, refers to the degree of confidence that the individuals have in themselves to maintain a new behavior when faced with challenging situations, and also relates to the balance of decisions, both favorable and contrasting factors in making changes [30].

The Transtheoretical Model was associated with the dialogic and problematizing pedagogy proposed by Paulo Freire in order to contribute to the design of actions capable of promoting the empowerment and autonomy of individuals. According to this approach, dialogue is the guiding thread for liberation, autonomy, and citizenship, being indispensable in the design of educational interventions [31]. 

For the development of the intervention, the participants were allocated into three groups according to the stages of change: pre-action (contemplation and pre-contemplation stages), preparation, and action (action and maintenance stages) [29,32,33]. For each of these groups, different change processes (cognitive and behavioral) were used to develop adequate and effective actions that facilitate behavior changes [26].

The themes of the educational actions were defined based on the previous qualitative investigation of practices, obstacles, facilitators, and social representations of FV consumption among the PAS users [34] and in the scientific literature [26]. The following themes were addressed: health and self-care; interfering factors in food choices; seasonality and cost of FV; preservation of the nutritional and sensorial quality of FV; guidelines for diversifying forms of preparation and consumption; consumption, portions and nutritional information of FV; and family support [26].

In all, 10 workshops were planned (structured work with groups focused around a central issue), and 4 actions in the environment (modifications in the environment with the insertion of unusual objects into the routine of the health service in order to promote reflection), repeated 540 and 171 times, respectively. A total of 4449 postcards in three formats were also distributed with eight different motivational messages aimed at the consumption of FV; a booklet informing about the importance of FV consumption, cleaning, purchase, and storage was also distributed in addition to healthy recipes. More details on the nutritional intervention are described in the article by Menezes et al. [26]. 

The average percentage of participants’ adherence to the nutritional intervention was 58.3%, with 24.3% having low adherence (30% or less participation), 26.5% having medium (31 to 70% participation), and 49.2% having high adherence (>70% participation) [35].

### 2.5. Outcome Variables 

The total energy (kcal) of the diet and the following macronutrients were investigated: carbohydrates, proteins, total fats, and fat subgroups (saturated, monounsaturated, polyunsaturated, trans, omega 3, and omega 6 linoleic acid). Macronutrients were analyzed according to their contribution of energy to the total energy value of the diet, and expressed as a percentage. The micronutrients investigated were vitamins A, B_1_, B_2_, B_3_, B_6_, B_12_, Folate, C, D, E, and the minerals were calcium, iron, phosphorus, magnesium, potassium, selenium, zinc, manganese, and sodium. To present the consumption of micronutrients and fiber, the measure of nutrient density was used, that is, the amount of nutrient or fiber consumed every 1000 kcal (g, mg, or µg/1000 kcal). 

The nutrients adequacy for the prevention and control of NCDs investigated were total, saturated, and trans fats; fibers; potassium; and sodium. Its adequacy was evaluated as follows: ≤30% energy intake from total fat; ≤10% energy intake from saturated fats; ≤1% energy intake from trans fats; ≥12.5 g/1000 kcal of dietary fiber; ≤1000 mg/1000 kcal for sodium; and ≥1755 mg/1000 kcal for potassium [5,6,7,8,9,10,11,12,13,14,15,16,17,18,19,20,21,22,23,24,25,26,27,28,29,30,31,32,33,34,35,36].

### 2.6. Covariables 

The sociodemographic variables investigated were age in years (median), years of education (median), sex (female and male), marital status (married/stable union; separated/divorced; single and widowed), and per capita family income (median). To calculate the per capita family income, the income of the residents of the household was added together and divided by the total number of residents.

The health variables investigated were health perception (poor/very bad, fair, or good/very good), regular physical activity (<150 min per week or ≥150 min per week), reported morbidity (hypertension and diabetes mellitus: yes or no), and BMI in kg/m^2^ (median and obesity: yes or no). The practice of physical activity was calculated from the weekly frequency and time spent from the activity questionnaire. Additionally, the time variable (in months) of user participation in the PAS (median) was evaluated. 

### 2.7. Data Analysis 

Very low (<500 kcal) or very high (>7000 kcal) total energy consumption values (*n* = 65) [37] were transformed into missing values, and added to those individuals who did not complete the R24h at baseline (*n* = 5) and/or in the reassessment (*n* = 1178). Then, imputation was performed.

The imputation of missing value was obtained from the round that had information. Therefore, those who did not have information in round 1 were imputed by round 2 and those who did not have information in round 2 were imputed by round 1. In this step, 10 possible values were generated from the imputation and we selected the average of these 10 values as the value imputed. When the individual had no value for either round 1 or round 2, the overall mean of observations over that period was used. And, when the imputation value was negative, the positive value was assigned.

Then, imputation was performed by the value of the observed round + sociodemographic variable (gender, age, and education). Finally, descriptive data analysis was undertaken and characterized to compare original and the imputed data.

After the data were imputed, descriptive statistical analyses were performed, with calculation of medians and interquartile ranges, and percentages to assess the sociodemographic and health characteristics of the participants. Differences between individuals in the IG and CG were evaluated using Pearson’s chi-square and Mann–Whitney’s statistical tests. 

Normality analyses were carried out. The Wilcoxon statistical test was used to assess the evolution of nutrient consumption of the IG and CG participants at baseline and at reassessment. To analyze the evolution of the adequacy of nutrient intake, the McNemar test was used.

The effectiveness of the collective intervention to encourage the consumption of FV was analyzed using the generalized estimation equation model (GEE) in order to verify whether the consumption of nutrients and the adequacy of their consumption, during the intervention, presented the same pattern of behavior according to intervention time and group (IG and CG). For the GEE analysis related to nutrient consumption, an unstructured working correlation matrix for continuous dependent variables was used, and the gamma distribution model with log linkage function was adopted due to the asymmetry of the variables. And the association measure used to present the results was the relative risk with a confidence interval of 95%. As for the GEE analyses, regarding the adequacy of nutrient intake, the interchangeable work correlation matrix was considered and the logistic distribution model was adopted, with the association measure used being the odds ratio with a confidence interval of 95%. Both GEE analyses were adjusted for the variables gender, age, years of study, time of participation in PAS, BMI, and nutrient consumption at baseline.

Data analyses were performed using the Data Analysis and Statistical Software (STATA) version 14.0. For the analysis of the GEE Model, the statistical program Statistical Package for the Social Sciences (SPSS) version 20.0 was used. In all analyses, a significance level lower than or equal to 5% was adopted (*p* value < 0.05). 

### 2.8. Ethical Standards Disclosure

This study was conducted according to the guidelines laid down in the Declaration of Helsinki and all procedures involving research study participants were approved by the Universidade Federal de Minas Gerais committee and Prefeitura Municipal de Belo Horizonte committee. A written informed consent was obtained from all subjects.

## 3. Results

A total of 3414 individuals participated in the study; 43.4% (*n* = 1428) allocated to the IG and 56.6% (*n* = 1931) to the CG. Most were women (88.1%), had a median of age of 58 (49–65) years, per capita family income of $301.33 (188.61–444.44), and education of 7 (4–11) years of study (Table 1). 

Prevalence of arterial hypertension was 53.2% (51.5–54.8) and of diabetes mellitus was 16.9% (15.7–18.2). Most participants practiced 150 min of physical activity or more per week (93.4%) and considered their health to be good/very good (71.7%). The median time of participation in the PAS was 16.7 months (7.0–30.5) and the BMI was 27.3 kg/m^2^ (24.4–30.6), with 28.9% (27.4–30.5) of the participants with obesity. Differences were identified between the IG and CG in relation to the medians of years of education, per capita family income, and time of participation in the PAS (Table 1).

When assessing the differences in nutrient intake between individuals in the IG and CG at baseline, lower energy (*p* = 0.04) and vitamin C (*p* = 0.01) consumption was observed among those in the IG in comparison to CG. The adequacy of nutrient intake for NDCs protection showed values below 50% adequacy for potassium (CG: 16.6%; and IG: 15.9%), trans fats (CG: 37.6%; and IG: 38.1%), and fibers (CG: 41.4%; and IG: 40.9%), without statistical differences between the groups (Table 2). 

When evaluating the effectiveness of the collective intervention to encourage the consumption of FV for nutrients intake, after adjustment, an inverse relationship was observed between participation in the intervention and the consumption of total energy [RR: 0.979 (0.960; 0.999)] and protein [RR: 0.978 (0.959; 0.997)] (Table 3). 

Analysis of the adequacy of the consumption of nutrients aiming at the prevention and control of NCDs showed that participants of both groups, after 12 months, reported an increase in the prevalence of adequacy of the consumption of total fat, saturated fats, fiber, and sodium, and reduced trans fats. Additionally, IG individuals showed an increase in the prevalence of adequacy of potassium intake. However, participation in the collective intervention to encourage the consumption of FV, after adjustment for possible confounding factors, was not associated with adequacy of nutrient intake for the prevention and control of NCDs (Table 4). 

## 4. Discussion

The collective intervention to encourage the consumption of FV was effective in reducing the consumption of energy and protein in the diet. In addition, after 12 months of follow-up, regardless of participation in the nutritional intervention, PAS participants showed a reduction in energy, omega 3, and sodium consumption, and an increase in the consumption of carbohydrates, fiber, vitamins A, B_1_, B_2_, B_3_, B_6_, B_12_, C, D, and E, folate, calcium, magnesium, potassium, selenium, manganese, zinc, and iron. IG individuals also showed a reduction in the consumption of proteins, and total, saturated and polyunsaturated fats; and an increase in monounsaturated fats consumption; while those in the CG reported an increase in phosphorus consumption. Regarding the evolution of the adequacy of the consumption of nutrients for the prevention and control of NCDs, improvements were observed for total fat, saturated, fiber, and sodium in both groups, and in potassium for IG; trans fats in worsened in both groups, with no differences related to participation in the nutritional intervention. 

Nutritional interventions are important tools to motivate and encourage healthy food choices, amplifying health service results. Systematic reviews that investigated the effect of nutritional interventions on the eating behavior of PHC users identified beneficial effects of interventions, such as increased consumption of FV and whole grains and reduced consumption of fatty foods, in addition to improving the nutrient profile of the diet [11,38]. However, these reviews point out the difficulty in evaluating the effectiveness of these interventions, either due to the methodological gaps in the studies [11,38] or due to the absence of markers and parameters that assess their effects on the prevention and control of NCDs [11]. In this sense, further studies on the subject are needed.

One of the eating behaviors that should be the target of nutritional interventions developed in PHC is the high consumption of energy and total fat, as it is related to excessive weight gain and the development and aggravation of NCDs [5]. This study showed a reduction in total dietary energy intake in both groups. In addition, participation in the collective intervention to encourage the consumption of FV, based on the Transtheoretical Model and on Paulo Freire’s dialogic and problematizing pedagogy, promoted an even greater reduction in energy consumption of the IG participants, in relation to the CG. Although the high prevalence of obesity among the study participants (28.9%) may have stimulated the search for food choices that promote an energy deficit [39], the results seem to reinforce the potential of health promotion services, such as the PAS, in the prevention and control of obesity, especially when they include, in their scope, actions to promote adequate and healthy food.

Regarding protein consumption, the median values of protein consumption at baseline remained above the recommendations for the prevention and control of NCDs (10% to 15% of the total energy value) [6]. After 12 months of follow-up, the consumption of this nutrient showed a significant reduction among the individuals in the IG, when compared to the CG, suggesting a possible trend towards the adequacy of this nutrient to follow the consumption recommendations through participation in the nutritional intervention. However, no effects of nutritional intervention on the consumption of most of the nutrients investigated were identified, including those aimed at the prevention and control of NCDs. This fact may be related to the monotony of FV consumption observed among PAS users [39].

It is essential to design and strengthen food and nutrition education actions within the scope of health services, such as the PAS, given its characteristics of health promotion and care. In addition, interventions aimed at improving the food environment around these health services are also necessary in order to support their clients in making healthier food choices. Actions developed in the health promotion services are strategic to promote adequate and healthy food choices. For example, longer participation in the PAS seems to be effective in improving the profile of food and nutrient consumption of its clients, as evidenced by a cross-sectional study that identified an increase in the consumption of home-prepared food and a reduction in UPF, as well as lower energy and fat consumption, and higher levels of carbohydrates, vitamin C, and calcium among those with longer time as participation in the health service [24]. A previous longitudinal study showed a reduction in the consumption of UPF and an increase in home-prepared food [23], possibly corroborating the results of this study, which showed an improvement in the consumption of nutrients over time, including those related to the prevention and control of NCDs, following the recommendations [5]. 

Understanding the role and importance of balanced consumption of nutrients for a healthy diet, and the prevention and control of NCDs, some parameters guide adequate nutrient consumption. It is noteworthy that our result which stated that a significant increase in fiber consumption and adequacy was observed for both groups (CG and IG) over the course of the study, is in line with the recommendation [5].

As for nutrients, the preference for unsaturated fats over saturated and trans fats [5] and limits for their consumption are recommended. A reduction in the consumption of total, saturated, and polyunsaturated fats was observed in this study; and an increase in monounsaturated fat was observed for the IG. In both groups, a reduction in the consumption of omega 6 and an improvement in the adequacy of consumption of total and saturated fats were also observed. Despite these positive results, a decline in the adequacy of trans fat intake was identified, which are part of an unhealthy diet, and therefore, should be avoided [4,5]. As a possible hypothesis for this negative result, the high consumption of UPF identified in this population is pointed out [23].

Regarding sodium, a consumption of less than 2 g per day is recommended, since its increased intake is associated with insufficient potassium consumption contributing to the development of arterial hypertension and increased risk of heart and cerebrovascular diseases [5]. In this study, there was a reduction in sodium consumption and an increase in potassium over time, as well as an improvement in the adequacy of sodium consumption for the total sample and potassium for the IG participants. It is noteworthy that these results may also be related to the high prevalence of individuals with arterial hypertension and diabetes mellitus participating in the study, since the presence of NCDs influences food choices [39]. But it is worth emphasizing that consumption decreased over time, which may show possible effects of the intervention capable of raising awareness of other changes in diet, in addition to the consumption of FV.

This study makes important contributions by demonstrating that health promotion service users tend to show improvement in the adequacy of nutrient consumption over time and that the collective intervention to encourage the consumption of FV, based on theories, potentiated these beneficial effects, further reducing total energy and protein consumption. It also showed greater adequacy in the consumption of nutrients, including those aimed at the prevention and control of NCDs, regardless of participation in the nutritional intervention, revealing the potential of the PAS to promote adequate and healthy eating. However, it is also vital that government strengthen public actions and policies that promote a healthy food environment consistent with the health recommendations offered by PAS. 

Among the limitations of this study is the possible information bias on food consumption, inherent to food surveys. To minimize their possible effects, the instruments were applied by a data collection team trained every six months and a home-measurement kit was used to help identify the portions of food consumed. Another limitation relates to the quality of the nutritional composition tables that contain restricted data in relation to the foods consumed by the population. To reduce this limitation, tables of the nutritional composition of foods commonly consumed in Brazil were used, developed from the Family Budget Surveys; food label information and/or recipe standardizations were added, when necessary. Additionally, not having investigated the intake of vitamin supplements and the fact that some study participants already had arterial hypertension or diabetes were some other limitations of the study, but adjustments were made to minimize this effect as well. 

The external validity of this study is limited, considering that the participants come from health promotion services, and may differ from the general population, including the motivation to make changes in health-related behaviors. However, its data are relevant for the Brazilian PHC and for the adequacy of public policies and educational actions developed within the scope of PAS, which includes, among its objectives, the prevention and control of NCDs. Finally, there was a loss of follow-up during the study; however, data imputation was performed to minimize possible interference from this loss. 

The strengths of this study are the sample size and the study design with a high level of scientific evidence, which allowed us to obtain reliable data on the effectiveness of nutritional intervention on the nutrient profile, including those aimed at the prevention and control of NCDs, gaps existing in the literature and identified by systematic reviews [11,38]. Furthermore, this study was conducted in a Brazilian PHC health service and demonstrated the potential of a collective intervention to encourage the consumption of FV to promote changes in the nutritional profile of the diet of its users, reinforcing the importance of carrying out nutritional interventions in health services and adapting them to local needs and national dietary guidelines in order to achieve effectiveness and sustainability. This study is rather important because it was carried out in a public health service and for the first time it verified, among users of the Brazilian unified health system, the effect of the intervention on the consumption of nutrients, including those related to the prevention and control of NCDs, which happen to be the main causes of morbidity and mortality in the world.

Results of this study pointed to the practical importance of health promotion services for the positive development of the nutrient consumption profile, including those associated with the prevention and control of NCDs. Thus, this study has practical importance, which can guide the expansion of actions carried out within the scope of PAS, including in its routine actions to promote adequate and healthy food.

## Figures and Tables

**Figure 1 ijerph-20-06267-f001:**
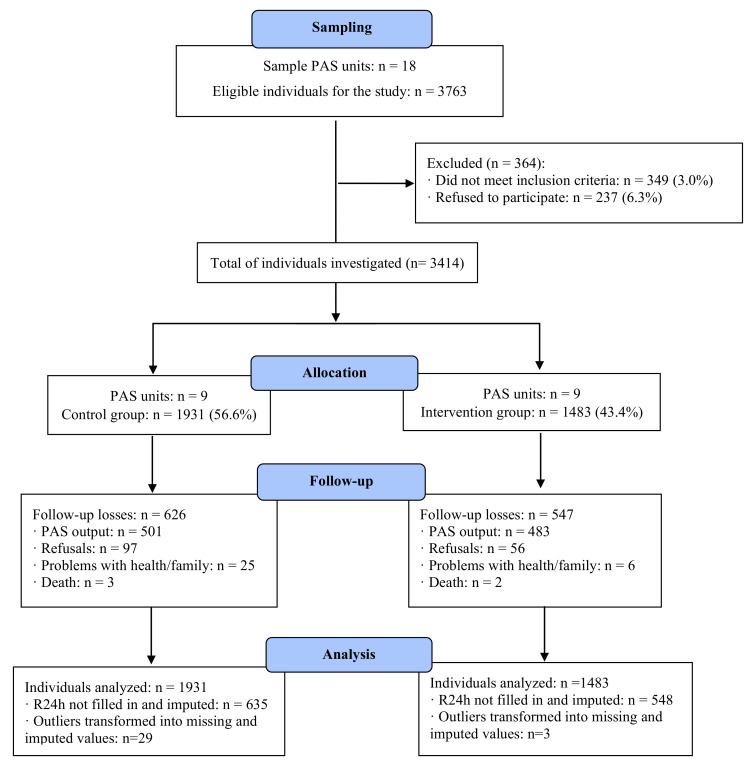
Study flowchart. Note: PAS: Health Academy Program. R24h: 24 h food recall.

**Table 1 ijerph-20-06267-t001:** Sociodemographic and health characteristics at baseline of Brazilian health promotion service users participating in a randomized controlled community trial. Belo Horizonte, Minas Gerais, Brazil (2013–2014).

Variables	Total Sample (*n* = 3414)		Baseline				*p* Value
			CG [*n* = 1931 (56.6%)]		IG [*n* = 1483 (43.4%)]		
	N	Values	*n*	Values	*n*	Values	
Gender, %							
Female	3007	88.1	1717	88.9	1290	87.0	0.08 *
Male	407	11.9	214	11.1	193	13.0	
Marital status ^1^, %							
Married	2102	61.6	1196	62.0	906	61.1	0.72 *
Separate	1311	38.4	734	38.0	577	38.9	
Age (years)	3414	58 (49–65)	1.931	58 (50–65)	1.483	57 (49–65)	0.10 ^#^
Years of study	3414	7 (4–11)	1.931	8 (4–11)	1.483	7 (4–11)	0.001 ^#^
Per capita family income ($) ^2^	3116	301.33 (188.61–444.44)	1.761	311.11 (200.00–488.89)	1355	301.33 (177.78–444.44)	0.001 ^#^
Self-perception of health ^1^							
Bad/Very bad	107	3.1	59	3.1	48	3.4	0.95 *
Fair	859	25.2	485	25.1	374	25.2	
Good/Very good	2447	71.7	1.386	71.8	1.061	71.5	
Physical activity ^3^							
<150 min per week	218	6.6	121	6.5	97	6.8	0.75 *
≥150 min per week	3058	93.4	1.731	93.5	1.327	93.2	
Time of participation in the PAS (months)	3414	16.7 (7.0–30.5)	1.931	17.8 (8.1–34.4)	1483	14.8 (5.8–27.2)	<0.001 ^#^
BMI ^4^ (kg/m^2^)	3264	27.3 (24.4–30.6)	1.849	27.3 (24.3–30.7)	1.415	27.3 (24.5–30.5)	0.83 ^#^
Diabetes mellitus ^5^, %	576	16.9 (15.7–18.2)	331	17.2 (15.5–18.9)	245	16.6 (14.8–18.6)	0.65 *
Arterial hypertension ^6^, %	1814	53.2 (51.5–54.8)	1.018	52.7 (50.5–55.0)	796	53.7 (51.2–56.2)	0.57 *
Obesity ^4^, %	944	28.9 (27.4–30.5)	539	29.1 (27.1–31.3)	405	28.6 (26.3–31.0)	0.74 *

Note: CG: control group. IG: intervention group; Values: percentage or median (P25–P75); BMI: body mass index. ^1^ Missing = 1. ^2^ Missing = 298. ^3^ Missing = 138. ^4^ Missing = 150. ^5^ Missing = 7. ^6^ Missing = 2. * Chi-square statistical test and ^#^ Mann–Whitney test.

**Table 2 ijerph-20-06267-t002:** Baseline characteristics related to nutrient intake and adequacy of Brazilian health promotion service users participating in a randomized controlled community trial. Belo Horizonte, Minas Gerais, Brazil (2013–2014).

Variables	Total Sample (*n* = 3414)	Baseline		*p* Value
		CG [*n* = 1931 (56.6%)]	IG [*n* = 1483 (43.4%)]	
	Values [Median (P_25_–P_75_)] ^1^	Values [Median (P_25_–P_75_)] ^1^	Values [Median (P_25_–P_75_)] ^1^	
Nutrients				
Total energy (kcal/d)	1354.2 (1061.1–1669.9)	1369.2 (1068.3–1691.1)	1328.6 (1046.6–1645.4)	0.04
Protein (%)	16.8 (14.1–19.9)	16.7 (14.1–19.8)	16.9 (14.1–20.2)	0.32
Carbohydrate (%)	53.8 (48.1–59.5)	54.0 (48.1–59.8)	53.6 (48.0–59.2)	0.28
Fiber (g/1000 kcal)	11.6 (9.0–14.7)	11.7 (9.1–14.8)	11.5 (8.9–14.7)	0.18
Total fat (%)	*29.8 (24.8–34.4)*	29.8 (24.7–34.3)	29.9 (24.9–34.6)	0.80
Saturated fat (%)	9.5 (7.4–11.6)	9.5 (7.3–11.6)	9.5 (7.4–11.8)	0.49
Monounsaturated fat (%)	9.6 (7.6–12.0)	9.7 (7.6–11.9)	9.6 (7.6–12.1)	0.86
Polyunsaturated fat (%)	5.9 (4.6–7.4)	5.9 (4.6–7.3)	6.0 (4.6–7.4)	0.43
Trans fat (%)	1.3 (0.8–2.2)	1.3 (0.8–2.2)	1.3 (0.8–2.2)	0.85
Omega 3 (%)	4.9 (3.8–6.1)	4.9 (3.8–6.1)	5.0 (3.9–6.2)	0.54
Omega 6 (%)	0.6 (0.5–0.9)	0.65 (0.5–0.9)	0.6 (0.5–0.9)	0.90
Vitamin A (mcg/1000 kcal)	316.1 (213.3–490.3)	314.4 (215.3–484.4)	319.1 (210.2–501.7)	0.43
Vitamin B_1_ (mg/1000 kcal)	0.7 (0.6–0.8)	0.7 (0.6–0.8)	0.7 (0.6–0.8)	0.75
Vitamin B_2_ (mg/1000 kcal)	1.1 (0.9–2.2)	1.1 (0.9–2.3)	1.1 (0.9–2.1)	0.90
Vitamin B_3_ (mg/1000 kcal)	7.5 (6.0–9.6)	7.5 (6.0–9.6)	7.5 (6.0–9.5)	0.67
Vitamin B_6_ (mg/1000 kcal)	0.9 (0.7–1.1)	0.9 (0.7–1.1)	0.92 (0.7–1.1)	0.85
Folate (mcg/1000 kcal)	140.5 (102.6–185.1)	141.1 (104.4–184.9)	139.8 (101.0–186.3)	0.50
Vitamin B_12_ (mcg/1000 kcal)	1.7 (1.2–2.5)	1.7 (1.2–2.5)	1.7 (1.2–2.5)	0.76
Vitamin C (mg/1000 kcal)	52.8 (22.9–100.1)	55.9 (23.9–104.1)	50.2 (21.6–95.4)	0.01
Vitamin D (mcg/1000 kcal)	1.6 (1.0–2.4)	1.6 (1.0–2.4)	1.6 (1.0–2.4)	0.52
Vitamin E (mg/1000 kcal)	2.4 (1.9–3.2)	2.5 (1.9–3.2)	2.4 (1.9–3.1)	0.11
Calcium (mg/1000 kcal)	343.4 (246.1–460.4)	347.7 (245.4–462.1)	336.8 (246.8–458.2)	0.60
Phosphorus (mg/1000 kcal)	552.3 (459.3–658.2)	550.3 (462.1–657.3)	556.1 (455.5–660.6)	0.69
Magnesium (mg/1000 kcal)	125.1 (104.2–151.3)	125.9 (104.8–151.7)	124.2 (103.1–149.7)	0.21
Potassium (mg/1000 kcal)	1348.8 (1114.4–1610.5)	1345.3 (1115.1–1616.4)	1348.8 (1111.2–1604.7)	0.73
Selenium (mcg/1000 kcal)	41.8 (31.5–54.3)	41.9 (31.4–54.4)	41.8 (31.7–54.1)	0.95
Sodium (mg/1000 kcal)	819.5 (622.2–1048.8)	822.7 (624.7–1055.3)	815.9 (619.7–1040.6)	0.20
Zinc (mg/1000 kcal)	5.2 (4.3–6.4)	5.2 (4.2–6.3)	5.3 (4.3–6.4)	0.12
Manganese (mg/1000 kcal)	1.3 (1.0–1.6)	1.3 (1.0–1.6)	1.3 (1.0–1.7)	0.73
Iron (mg/1000 kcal)	4.9 (4.1–5.9)	4.97 (4.1–5.9)	5.0 (4.2–5.9)	0.47
Prevalence of nutrient adequacy				
Total fat [% (95%CI)] ^2^	51.7 (49.9–53.3)	51.7 (49.4–53.9)	51.6 (49.1–54.2)	0.51
Saturated fat [% (95%CI)] ^2^	56.8 (55.2–58.5)	57.4 (55.1–59.6)	56.2 (53.6–58.7)	0.25
Trans fat [% (95%CI)] ^2^	37.8 (36.2–39.4)	37.6 (35.5–39.8)	38.1 (35.7–40.6)	0.49
Fibers [% (95%CI)] ^2^	41.2 (39.6–42.9)	41.4 (39.2–43.6)	40.9 (38.4–43.4)	0.40
Sodium [% (95%CI)] ^2^	70.7 (69.2–72.2)	70.0 (67.9–72.0)	71.7 (69.4–74.0)	0.14
Potassium [% (95%CI)] ^2^	16.3 (15.1–17.6)	16.6 (15.0–18.3)	15.9 (14.1–17.9)	0.32

Note: CG: control group. IG: intervention group. The calculation of the adequacy prevalence considered recommendations for the prevention of NCDs recommended by the WHO (WHO, 2003, 2013, 2018). ^1^ Mann–Whitney test. ^2^ Chi-square test.

**Table 3 ijerph-20-06267-t003:** Effectiveness of collective intervention to encourage the consumption of FV for nutrients intake among users of Brazilian health promotion services. Belo Horizonte, Minas Gerais, Brazil (2013–2014).

Variables	CG [Median (P_25_–P_75_)] (*n* = 1931)			IG [Median (P_25_–P_75_)] (*n* = 1483)			RR (95%CI)	RR ** (95%CI)
	Baseline	Reassessment	*p* Value ^1^	Baseline	Reassessment	*p* Value ^1^		
Total energy (kcal/d)	1369.2 (1068.3–1691.1)	1279.0 (1072.9–1554.2)	<0.001	1328.6 (1046.6–1645.4)	1241.9 (1017.3–1473.3)	<0.001	0.981 (0.959; 1.003)	0.979 (0.960; 0.999)
Protein (%)	16.7 (14.1–19.8)	16.8 (14.9–19.0)	0.24	16.9 (14.1–20.2)	16.9 (14.9–18.9)	0.002	0.978 (0.959; 0.997)	0.978 (0.959; 0.997)
Carbohydrate (%)	54.0 (48.1–59.8)	54.5 (50.2–58.8)	0.003	53.6 (48.0–59.2)	54.5 (50.8–58.6)	<0.001	1.011 (0.999; 1.024)	1.010 (0.996; 1.023)
Fiber *	11.7 (9.1–14.8)	12.5 (10.2–15.1)	<0.001	11.5 (8.9–14.7)	12.6 (10.4–15.1)	<0.001	1.018 (0.993; 1.043)	1.013 (0.988; 1.039)
Total fat (%)	29.8 (24.7–34.3)	29.5 (26.3–32.8)	0.14	29.9 (24.9–34.6)	29.4 (26.0–32.6)	0.003	0.990 (0.973; 1.008)	0.992 (0.974; 1.011)
Saturated fat (%)	9.5 (7.3–11.6)	9.5 (8.1–10.8)	0.66	9.5 (7.4–11.8)	9.5 (8.0–10.8)	0.03	0.994 (0.968; 1.020)	0.991 (0.964; 1.018)
Monounsaturated fat (%)	9.7 (7.6–11.9)	9.8 (8.4–11.3)	0.86	9.6 (7.6–12.1)	9.7 (8.3–11.1)	0.04	0.986 (0.961; 1.012)	0.992 (0.966; 1.018)
Polyunsaturated fat (%)	5.9 (4.6–7.3)	5.9 (4.9–6.8)	0.08	6.0 (4.6–7.4)	5.8 (5.0–6.7)	<0.001	0.974 (0.948; 1.001)	0.980 (0.954; 1.007)
Trans fat (%)	1.3 (0.8–2.2)	1.3 (0.9–2.0)	0.30	1.3 (0.8–2.2)	1.4 (0.9–2.1)	0.76	0.989 (0.936; 1.045)	1.021 (0.969; 1.076)
Omega 3 (%)	4.9 (3.8–6.1)	4.8 (3.9–5.9)	0.01	5.0 (3.9–6.2)	4.9 (3.9–5.9)	<0.001	0.983 (0.953; 1.013)	0.986 (0.958; 1.015)
Omega 6 (%)	0.65 (0.51–0.87)	0.67 (0.51–0.86)	0.13	0.65 (0.51–0.86)	0.69 (0.52–0.88)	0.90	1.030 (0.950; 1.116)	1.029 (0.973; 1.088)
Vitamin A *	314.4 (215.3–484.4)	384.1 (244.4–607.2)	<0.001	319.1 (210.2–501.7)	396.9 (242.0–639.0)	<0.001	0.915 (0.795; 1.054)	1.007 (0.909; 1.115)
Vitamin B_1_ *	0.68 (0.57–0.81)	0.71 (0.60–0.83)	<0.001	0.68 (0.57–0.80)	0.71 (0.61–0.82)	<0.001	1.004 (0.983; 1.024)	1.000 (0.979; 1.022)
Vitamin B_2_ *	1.1 (0.9–2.3)	1.5 (1.0–8.3)	<0.001	1.1 (0.9–2.1)	1.7 (0.9–8.3)	<0.001	1.117 (0.926; 1.349)	1.060 (0.914; 1.230)
Vitamin B_3_ *	7.5 (6.0–9.6)	8.0 (6.6–9.6)	0.017	7.5 (6.0–9.5)	8.0 (6.5–9.7)	0.01	1.005 (0.971; 1.040)	1.009 (0.978; 1.042)
Vitamin B_6_ *	0.93 (0.75–1.14)	0.98 (0.78–1.18)	0.001	0.92 (0.75–1.15)	0.98 (0.80–1.21)	<0.001	1.029 (0.952; 1.113)	1.019 (0.977; 1.062)
Folate *	141.1 (104.4–184.9)	156.7 (124.7–194.9)	<0.001	139.8 (101.0–186.3)	160.5 (124.9–199.6)	<0.001	1.017 (0.982; 1.053)	1.015 (0.981; 1.050)
Vitamin B_12_ *	1.7 (1.2–2.5)	2.1 (1.2–3.2)	<0.001	1.7 (1.2–2.5)	2.0 (1.2–3.5)	<0.001	0.900 (0.751; 1.079)	1.023 (0.907; 1.154)
Vitamin C *	55.9 (23.9–104.1)	78.1 (41.0–117.7)	<0.001	50.2 (21.6–95.4)	77.5 (41.7–120.4)	<0.001	1.090 (0.992; 1.199)	1.054 (0.976; 1.138)
Vitamin D *	1.6 (1.0–2.4)	1.7 (0.8–3.1)	<0.001	1.6 (1.0–2.4)	1.6 (0.7–3.1)	0.02	0.848 (0.666; 1.081)	0.835 (0.641; 1.087)
Vitamin E *	2.5 (1.9–3.2)	2.8 (2.1–3.6)	<0.001	2.4 (1.9–3.1)	2.8 (2.1–3.7)	<0.001	1.057 (1.001; 1.116)	1.020 (0.970; 1.072)
Calcium *	347.7 (245.4–462.1)	364.4 (272.3–464.6)	0.001	336.8 (246.8–458.2)	356.3 (268.5–455.1)	0.03	0.982 (0.953; 1.011)	0.987 (0.958; 1.017)
Phosphorus *	550.3 (462.1–657.3)	562.8 (475.2–661.9)	0.04	556.1 (455.5–660.6)	557.6 (472.2–658.1)	0.75	0.977 (0.956; 0.998)	0.981 (0.960; 1.002)
Magnesium *	125.9 (104.8–151.7)	129.4 (110.3–151.5)	0.04	124.2 (103.1–149.7)	130.1 (109.8–153.4)	<0.001	1.008 (0.988; 1.028)	1.005 (0.986; 1.025)
Potassium *	1345.3 (1115.1–1616.4)	1413.2 (1205.3–1639.7)	<0.001	1348.8 (1111.2–1604.7)	1414.1 (1183.8–1657.4)	<0.001	1.006 (0.988; 1.025)	1.002 (0.983; 1.021)
Selenium *	41.9 (31.4–54.4)	44.9 (34.1–58.4)	<0.001	41.8 (31.7–54.1)	44.3 (32.9–57.6)	<0.001	0.964 (0.915; 1.016)	0.952 (0.906; 1.001)
Sodium *	822.7 (624.7–1055.3)	781.3 (621.0–961.8)	<0.001	815.9 (619.7–1040.6)	778.6 (603.0–956.3)	<0.001	1.011 (0.975; 1.048)	0.995 (0.963; 1.028)
Zinc *	5.2 (4.2–6.3)	5.4 (4.5–6.4)	<0.001	5.3 (4.3–6.4)	5.4 (4.5–6.4)	0.04	0.986 (0.961; 1.011)	0.981 (0.957; 1.006)
Manganese *	1.3 (1.0–1.6)	1.4 (1.0–2.3)	<0.001	1.3 (1.0–1.7)	1.5 (1.0–2.9)	<0.001	0.964 (0.699; 1.329)	1.151 (0.957; 1.384)
Iron *	4.97 (4.1–5.9)	5.1 (4.3–5.9)	0.03	5.0 (4.2–5.9)	5.1 (4.4–6.1)	<0.001	1.010 (0.987; 1.034)	1.006 (0.983; 1.028)

Note: CG: control group. IG: intervention group. RR: Relative risk. *: mg/1000 kcal. RR: relative risk from generalized estimating equations. ^1^ Wilcoxon test. ** Adjusted for gender, age, years of education, time of participation in the PAS, body mass index and nutrient consumption at baseline.

**Table 4 ijerph-20-06267-t004:** Effectiveness of collective intervention to encourage the consumption of FV on the adequacy of nutrient consumption for the prevention of NCDs among users of the Brazilian health promotion service. Belo Horizonte, Minas Gerais, Brazil (2013–2014).

Variables	CG [% (IC95%)] (*n* = 1931)		*p* Value ^1^	IG [% (IC95%)] (*n* = 1483)		*p* Value ^1^	OR (95%CI) ^2^	OR (95%CI) *^,2^
	Baseline	Reassessment		Baseline	Reassessment			
Total fat	51.7 (49.4–53.9)	54.7 (52.5–56.9)	0.03	51.6 (49.1–54.2)	55.9 (53.4–58.4)	0.007	1.05 (0.89–1.24)	1.05 (0.88–1.24)
Saturated fat	57.4 (55.1–59.6)	61.3 (59.1–63.5)	0.005	56.2 (53.6–58.7)	60.7 (58.2–63.1)	0.005	1.02 (0.86–1.21)	0.99 (0.83–1.19)
Trans fat	37.6 (35.5–39.8)	31.4 (29.3–33.5)	<0.001	38.1 (35.7–40.6)	30.3 (28.0–32.7)	<0.001	0.93 (0.78–1.11)	0.93 (0.77–1.11)
Fibers	41.4 (39.2–43.6)	49.8 (47.6–52.0)	<0.001	40.9 (38.4–43.4)	50.8 (48.3–53.4)	<0.001	1.06 (0.90–1.25)	1.06 (0.90–1.26)
Sodium	70.0 (67.9–72.0)	79.3 (77.5–81.1)	<0.001	71.7 (69.4–74.0)	79.8 (77.7–81.8)	<0.001	0.95 (0.77–1.16)	0.93 (0.75–1.16)
Potassium	16.6 (15.0–18.3)	16.4 (14.8–18.1)	0.88	15.9 (14.1–17.9)	18.2 (16.3–20.2)	0.04	1.19 (0.96–1.48)	1.15 (0.91–1.43)

Note: CG: control group. IG: intervention group. OR: odds ratio. The calculation of the adequacy prevalence considered recommendations for the prevention of NCDs recommended by the WHO (WHO, 2003, 2013, and 2018). ^1^ Mc-Nemar. ^2^ generalized estimating equations. * Adjusted for gender, age, years of education, time of participation in the PAS, body mass index and nutrient consumption at baseline.

## Data Availability

Contact the corresponding author.

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
