# Peer review of "Evaluation of Nutrient Consumption for the Prevention of Chronic Diseases in Health Promotion Services: A Controlled and Randomized Community Trial to Promote Fruits and Vegetables"

_ijerph, 2023, doi:10.3390/ijerph20136267_

Round 1

Reviewer 1 Report

Please justify all references text

The text sounds good and enough.

Author Response

  • Reviewer: 1

  1. Please justify all references text

Thank you for your suggestions.

Reviewer 2 Report

1.       This study focuses on fruits and vegetables (FV). Hence the author could rewrite the tittle based on the aim.

2.       The participants are not clear. The author has to identify the participant's condition. Are they sick or healthy?

3.       Did the author calculate the normality?

4.       In this study, how the randomization was performed?

5.       This study, the inclusion and exclusion criteria are not clear.

6.       the author has to explain the practical application of this study in the discussion section.

7.       Describe the limitation and straightness and suggestions for future study.

8.       What is the home messages?

It is ok

Author Response

  • Reviewer: 2
  1. This study focuses on fruits and vegetables (FV). Hence the author could rewrite the tittle based on the aim.

Thanks for the suggestion. The intervention proposed on this study focuses on fruits and vegetables, but the evaluated consumption was of all meals. But we rewritten the title to clarify this point, according to the suggestion: "Evaluation of Nutrient Consumption for the Prevention of Chronic Diseases in Health Promotion Services: A Controlled and Randomized Community Trial to promote Fruit and Vege-table"..

  1. The participants are not clear. The author has to identify the participant's condition. Are they sick or healthy?

Thank you for putting this perspective. The study investigates all PAS participants, thus including both healthy and sick individuals. We add this information in the methods. Table 1 presents the prevalence’s.

  1. Did the author calculate the normality?

Yes, normality analyzes were conducted. We review the methods to clarify this question.

  1. In this study, how the randomization was performed?

Thank you for the question. We used simple random sampling. We have rewritten sentence.

  1. This study, the inclusion and exclusion criteria are not clear.

Thank you for the suggestion. We have rewritten the paragraph for this purpose.

  1. the author has to explain the practical application of this study in the discussion section.

Thank you for the suggestion. We agree. We have revised the last paragraph of the discussion for this purpose.

  1. Describe the limitation and straightness and suggestions for future study.

Thank you for the suggestion. We agree. We have revised the discussion.

  1. What is the home messages?

They were presented in the first paragraph of the discussion.

Reviewer 3 Report

The manuscript captures an interesting (but not unique) approach to addressing NCDs. A major issue for the authors to clarify is whether their focus is on disease prevention or management or both. In certain instances, they mentioned the prevention of NCDs, but they also described the "prevention and control of NCDs." Additionally, it may be useful for the authors to clearly state the unique contribution of their work to the current body of evidence. There are many studies that have looked at increased FV consumption with respect to the prevention/management of NCDs. What further knowledge is this particular work contributing? Some more detailed comments on the manuscript are below:

Introduction

·       Page 2, line 44: I believe the authors meant to write “It”, not Its at the beginning of the page

Methods

·       Page 2, lines 69-70: This sentence is unclear. Do the authors mean that the program provided free exercise sessions?

·       Page 2, line 78: One of the words does not appear clearly in the text, but it seems like the authors meant to write “in operation.” Can they confirm?

·       Page 3: Figure 1 is quite helpful. One recommendation is for the authors to consider writing “total number of study participants” instead of total of individuals investigated. Also, can the authors check their figures or provide some clarity? The number of people who refused to participate and the number of people who did not meet inclusion criteria gives a sum of 349, not 364.

·       Page 6, line 241: The authors stated, “Finally, was carried out and characterized by descriptive data analysis.” Can they clarify what this sentence means? Also, the description of the imputation of food consumption on page 6 is confusing and not easy to follow.

·       Page 7, line 262: I think the authors meant to write “GEE analysis” not GEE analyzes. The word also needs to be corrected in subsequent text.

Results

·       Page 7, Table 1: It looks like the table has 1.717 for the number of female participants in the control group. I assume this should be 1717? The authors should double-check other variables in the table to confirm all figures are correct.

·       Pages 9-11: The authors should consider if there may be a more succinct way of presenting the information in Table 3.

There are some grammatical errors and a few other issues that can be fixed. Some of these have been highlighted in my previous comments.

Author Response

  • Reviewer: 3

The manuscript captures an interesting (but not unique) approach to addressing NCDs. A major issue for the authors to clarify is whether their focus is on disease prevention or management or both. In certain instances, they mentioned the prevention of NCDs, but they also described the "prevention and control of NCDs." Additionally, it may be useful for the authors to clearly state the unique contribution of their work to the current body of evidence. There are many studies that have looked at increased FV consumption with respect to the prevention/management of NCDs. What further knowledge is this particular work contributing? Some more detailed comments on the manuscript are below:

The authors would like to thank you for very insightful comment. They have helped to improve the quality of the or work. As suggested by one of the reviewers the entire text was aligned with prevention and control.

Regarding additional knowledge this particular work is contributing. This study, with a high level of scientific evidence, which allowed us to obtain reliable data on the effectiveness of nutritional intervention on the nutrient profile, including those aimed at the prevention and control of NCDs, gaps existing in the literature and identified by systematic reviews. Furthermore, this study was conducted in a Brazilian PHC health service and demon-strated the potential of a collective intervention to encourage the consumption of FV to promote changes in the nutritional profile of the diet of its users, reinforcing the im-portance of carrying out nutritional interventions in health services and adapting them to local needs and national dietary guidelines in order to achieve effectiveness and sustaina-bility. This study is unprecedented in being conducted in a public health service and veri-fying the effect of intervention on the consumption of nutrients, including those related to the prevention and control of NCDs, the main causes of morbidity and mortality in the world. And its results pointed to the practical importance of health promotion services for the positive evolution of the nutrient consumption profile, including those associated with the prevention and control of NCDs. In this way, the practical importance of the findings of this study is highlighted, which can guide the expansion of the scope of ac-tions carried out within the scope of the PAS, including in its routine actions to promote adequate and healthy food. Essa relevancia foi apresentada na discussão.

Introduction

- Page 2, line 44: I believe the authors meant to write “It”, not Its at the beginning of the page

We have changed it. Thank you.

Methods

- Page 2, lines 69-70: This sentence is unclear. Do the authors mean that the program provided free exercise sessions?

Yes. Thank you for the suggestion. We have rewritten the sentence.

-  Page 2, line 78: One of the words does not appear clearly in the text, but it seems like the authors meant to write “in operation.” Can they confirm?

Yes. Thank you for the suggestion. We have rewritten the word.

- Page 3: Figure 1 is quite helpful. One recommendation is for the authors to consider writing “total number of study participants” instead of total of individuals investigated. Also, can the authors check their figures or provide some clarity? The number of people who refused to participate and the number of people who did not meet inclusion criteria gives a sum of 349, not 364.

Thank you. We would like to apologize for the error. the entire figure was carefully revised and the total number of participants added.

  • Page 6, line 241: The authors stated, “Finally, was carried out and characterized by descriptive data analysis.” Can they clarify what this sentence means? Also, the description of the imputation of food consumption on page 6 is confusing and not easy to follow.

Thank you for your question. This sentence has been rewritten to make it clearer that descriptive data analysis was conducted to compare original and the imputed data. With regard to imputation, we agree that this is challenging. We revised the text to reduce text and make this issue clearer: “[…] Very low (< 500 kcal) or very high (> 7000 kcal) total energy consumption values (n = 65) [39] were transformed into missing values, and added to those individuals who did not complete the R24h at baseline (n = 5) and/or in the reassessment (n = 1,178). Then, imputation was performed. The imputation of missing value was carried from the information of the round that had information. Therefore, those who did not have information in round 1 were imputed by round 2 and those who did not have information in round 2 were imputed by round 1. In this step, 10 possible values were generated from the imputation and we selected the average of these 10 values as the value imputed. When the individual had no value for either round 1 or round 2, the overall mean of observations over that period was used. And, when the imputation value was negative, the positive value was assigned. Then, imputation was performed by the value of the observed round + sociodemo-graphic variable (gender, age, education). Finally, was carried out and characterized by descriptive data analysis to compare original and the imputed data.[…]”

  • Page 7, line 262: I think the authors meant to write “GEE analysis” not GEE analyzes. The word also needs to be corrected in subsequent text.

Fixed. Thank you.

Results

- Page 7, Table 1: It looks like the table has 1.717 for the number of female participants in the control group. I assume this should be 1717? The authors should double-check other variables in the table to confirm all figures are correct.

 Fixed. Thank you.

- Pages 9-11: The authors should consider if there may be a more succinct way of presenting the information in Table 3.

Yes, revised. We decided to focus on the results of the effect of the intervention

Reviewer 4 Report

In the article ‘Evolution of Nutrient Consumption for the Prevention of Chronic Diseases in Health Promotion Services: A Controlled and Randomized Community Trial’, it is a valuable study in the public health field. However, there are some critical questions.

1.     I guess that the title may be ‘Evaluation of Nutrient Consumption for the Prevention of Chronic Diseases in Health Promotion Services: A Controlled and Randomized Community Trial’.

2.     In the line 5-6: …keratin-8 and keratin-18, forming heterodimers expressed in hepatocytes [6]. Abstract: To evaluate the effectiveness of a collective intervention to encourage the consumption of fruits and vegetables on the nutrients intake for the prevention of chronic non-communicable diseases (NCDs). This sentence has grammar and structure errors.

3.     In the line 8-10: While the individuals in the Control Group (CG) participated in regular physical exercise, those in the Intervention Group (IG) also participated in a collective intervention.. Please delete one punctuation mark.

In the article ‘Evolution of Nutrient Consumption for the Prevention of Chronic Diseases in Health Promotion Services: A Controlled and Randomized Community Trial’, it is a valuable study in the public health field. However, there are some critical questions.

1.     I guess that the title may be ‘Evaluation of Nutrient Consumption for the Prevention of Chronic Diseases in Health Promotion Services: A Controlled and Randomized Community Trial’.

2.     In the line 5-6: …keratin-8 and keratin-18, forming heterodimers expressed in hepatocytes [6]. Abstract: To evaluate the effectiveness of a collective intervention to encourage the consumption of fruits and vegetables on the nutrients intake for the prevention of chronic non-communicable diseases (NCDs). This sentence has grammar and structure errors.

3.     In the line 8-10: While the individuals in the Control Group (CG) participated in regular physical exercise, those in the Intervention Group (IG) also participated in a collective intervention.. Please delete one punctuation mark.

Author Response

  • Reviewer: 4

In the article ‘Evolution of Nutrient Consumption for the Prevention of Chronic Diseases in Health Promotion Services: A Controlled and Randomized Community Trial’, it is a valuable study in the public health field. However, there are some critical questions.

Thank you for your suggestions. The whole manuscript was carefully reviewed and restructured in order to better connect the text and help the readers with a clear and concise text. We provided more information and expanded some points to clarify, considering all the questions.

  1. I guess that the title may be ‘Evaluation of Nutrient Consumption for the Prevention of Chronic Diseases in Health Promotion Services: A Controlled and Randomized Community Trial’.

Agreed, thank you. We have rewritten the title.

  1. In the line 5-6: …keratin-8 and keratin-18, forming heterodimers expressed in hepatocytes [6]. Abstract: To evaluate the effectiveness of a collective intervention to encourage the consumption of fruits and vegetables on the nutrients intake for the prevention of chronic non-communicable diseases (NCDs). This sentence has grammar and structure errors.

Thanks for your comment. All text was proofread by a native speaker and such gamma errors resolved.

  1. In the line 8-10: While the individuals in the Control Group (CG) participated in regular physical exercise, those in the Intervention Group (IG) also participated in a collective intervention. Please delete one punctuation mark.

Fixed. Thank you.

Round 2

Reviewer 2 Report

It is okay.

Author Response

Thank you for your review.

Reviewer 3 Report

Thanks for incorporating the suggestions.

For line 327, I believe the authors meant to state that "Normality analyses were carried out."

Also, there are still some issues with the n and N values in Table 1. Please double-check the data being presented.

If the authors are using "years of schooling" and "years of education" to mean the same thing, I suggest that they become consistent by using just one term throughout the manuscript. 

Lines 562-563: The authors stated, "In this study, a significant increase in fiber consumption and adequacy was observed for both groups (CG and IG) over the course of the study." They should consider revising this statement to: A significant increase in fiber consumption and adequacy was observed for both groups (CG and IG) over the course of this study.

Lines 666-669-The following statement is unclear, "This study is unprecedented in being conducted in a public health service and 666 verifying the effect of intervention on the consumption of nutrients, including those related to the prevention and control of NCDs, the main causes of morbidity and mortality 668 in the world."

Many improvements have been made, but there is still some room for an even stronger paper. In the previous section for comments, I highlighted a few places where changes can be made.

Author Response

  • For line 327, I believe the authors meant to state that "Normality analyses were carried out."

Thank you for the suggestion. We have rewritten the sentence.

  • Also, there are still some issues with the n and N values in Table 1. Please double-check the data being presented.

We have changed it. Thank you.

  • If the authors are using "years of schooling" and "years of education" to mean the same thing, I suggest that they become consistent by using just one term throughout the manuscript. Yes. Thank you for the suggestion. We have changed it.

  • Lines 562-563: The authors stated, "In this study, a significant increase in fiber consumption and adequacy was observed for both groups (CG and IG) over the course of the study." They should consider revising this statement to: A significant increase in fiber consumption and adequacy was observed for both groups (CG and IG) over the course of this study.

Thank you for the suggestion. We have rewritten the sentence.

  • Lines 666-669-The following statement is unclear, "This study is unprecedented in being conducted in a public health service and 666 verifying the effect of intervention on the consumption of nutrients, including those related to the prevention and control of NCDs, the main causes of morbidity and mortality 668 in the world."

Yes. Thank you for the suggestion. We have rewritten the sentence.

Comments on the Quality of English Language

  • Many improvements have been made, but there is still some room for an even stronger paper. In the previous section for comments, I highlighted a few places where changes can be made.

Thanks for your comment. All text was proofread by a native speaker and such gamma errors resolved.